# A Long-Term Comparative Analysis of Endovascular Coiling and Clipping for Ruptured Cerebral Aneurysms: An Individual Patient-Level Meta-Analysis Assessing Rerupture Rates

**DOI:** 10.3390/jcm13061778

**Published:** 2024-03-20

**Authors:** Johannes Wach, Martin Vychopen, Agi Güresir, Alexandru Guranda, Ulf Nestler, Erdem Güresir

**Affiliations:** Department of Neurosurgery, University Hospital Leipzig, 04103 Leipzig, Germany; martin.vychopen@medizin.uni-leipzig.de (M.V.); alexandru.guranda@medizin.uni-leipzig.de (A.G.); ulf.nestler@medizin.uni-leipzig.de (U.N.); erdem.gueresir@medizin.uni-leipzig.de (E.G.)

**Keywords:** aneurysm, clipping, coiling, individual patient data, long-term outcome, meta-analysis, rerupture

## Abstract

**Background:** Although the initial functional outcome findings of the International Subarachnoid Aneurysm Trial (ISAT) study favored coiling at one year after aneurysmal subarachnoid hemorrhage (aSAH), concerns arose regarding limited long-term rerupture data. This meta-analysis is the first to analyze longitudinal individual patient data (IPD) of target aneurysm rerupture in terms of treatment modality. **Methods:** The present meta-analysis included studies that compared clipping with coiling of ruptured aneurysms regarding long-term rerupture. Rerupture rates’ individual patient data (IPD) were extracted from published Kaplan–Meier curves utilizing the R package IPDfromKM in R Version 4.3.1. **Results**: A total of 3153 patients from two studies were included. The clipping arm included 1755 patients, whereas the coiling arm included 1398 patients. Median reconstructed follow-up was 6.1 years (IQR = 0.5–11.7). The rerupture rates in the clipping arm and the coiling arm were 0.5% and 1.5%, respectively (*p* = 0.002). Kaplan–Meier chart analysis of the 3153 patients revealed a shortened time to rerupture in the coiling arm (log-rank test: *p* = 0.01). The hazard ratio (HR) for coiling compared with clipping regarding rerupture was 3.62 (95% CI:1.21–10.86, *p* = 0.02). **Conclusion:** Target aneurysm rerupture was rare beyond the initial year. Pooled long-term IPD from the 3153 patients revealed that reruptures of target aneurysms are more common after coiling and might be considered in the pretherapeutic decision-making process for aSAH.

## 1. Introduction

Aneurysmal subarachnoid hemorrhage (aSAH) is a critical condition characterized by considerable morbidity and a 1-month mortality rate of 21%, despite recent advancements in treatment strategies [1,2]. Complete obliteration of the ruptured aneurysm dome remains the therapy of choice [3]. According to the 2023 Guidelines for the Management of the Patients with aSAH, clipping or coiling is recommended where clinically feasible [4].

In acute phase and in mid-term follow-up, the results of both recommended procedures show no significant difference if indicated properly [3,5]. However, rerupture with subsequent second hemorrhage event presents a dreaded long-term complication with possibly disastrous outcome and should be prevented at all costs [6]. A potential explanation for the rebleeding event might be an aneurysm remnant, which grows over the course of time and eventually causes a second aSAH [7]. Long-term follow-up is therefore needed to reliably assess rerupture rates and accurately compare both occlusion methods.

For such analysis, individual patient data (IPD) meta-analysis seems to be the most appropriate method to identify the timepoints with increased risk of rerupture, which are necessary to determine the optimum follow-up interval scheduling or retreatment planning [8]. To date, IPD meta-analysis of rebleeding rates using longitudinal time-to-event data has not been performed.

The aim of our study is to conduct an IPD meta-analysis of the rebleeding rates in patients with aSAH who underwent either surgical clipping or endovascular coiling and were followed-up and reported on accordingly [8].

## 2. Materials and Methods

The systematic review and meta-analysis were conducted according to the Preferred Reporting Items for Systematic Reviews and Meta-Analyses (PRISMA, see Appendix A) statement for IPD development cohorts, and the study protocol was prospectively registered in the “International Prospective Register of Systematic Reviews” (PROSPERO, Registration ID: CRD42023463253) in 2023 [9]. The detailed prespecified protocol is available in the PROSPERO register using this ID.

### 2.1. Search Strategy and Study Inclusion Criteria

We searched for “aneurysmal subarachnoid hemorrhage rerupture” in PubMed, Medline, Cochrane, and Embase databases until 1 August 2023 and found 240 eligible studies. We performed title screening, abstract screening, and in case of uncertainty a whole text screening to search for eligible studies. Subsequently, we screened meta-analyses on subarachnoid hemorrhage to further identify possibly eligible studies. Included were all studies describing longitudinal rerupture-free survival data of initially treated ruptured aneurysms (“target aneurysm”) after coiling or lipping with Kaplan–Meier charts and corresponding patient number at risk tables. Studies reporting only rates without longitudinal individual patient data displayed in a Kaplan–Meier chart with number at risk tables were not eligible for this individual patient data meta-analysis. We excluded case studies, study protocols, non-clinical trials, and meta-analyses. The rerupture event was limited to rebleeding from a previously ruptured aneurysm that had been successfully treated using coiling or clipping. No data on rerupture from untreated aneurysms at other sites or de novo aneurysms that caused a second aSAH were included.

Two reviewers (JW and MV) independently screened abstracts and full-text articles for two rounds, with any remaining conflicts resolved by a third reviewer (EG).

### 2.2. Quality Assessment

The National Institutes of Health Quality Assessment Tool for observational cohort and cross-sectional studies (NIH-QAT) was used to evaluate the quality and risk of bias in the included studies [10].

### 2.3. Data Extraction

Two authors (JW and MV) independently extracted the following data from the studies: clinical and imaging characteristics of aSAH patients, prevalence of coiled or clipped ruptured cerebral aneurysms, and data regarding rerupture-free survival in cases with coiled or clipped ruptured cerebral aneurysms. The IPD information of rerupture-free survival was extracted from the published Kaplan–Meier plots and patient number at risk tables using Digitizelt (Version 2.5.10 for macOS, Braunschweig, Germany) [11]. This method was performed for each individual subgroup of aSAH patients who underwent coiling or clipping. The extracted rerupture-free survival data and the published number at risk tables were used to reconstruct the Kaplan–Meier curves for the individual studies using the method of Liu et al. with the R package IPDfromKM (Version 0.1.10) in R studio (Rstudio, Boston, MA, USA) and R (Version 4.3.1) [12]. Furthermore, the number at risk tables were also created. Afterwards, the reconstructed curves, number at risk tables, estimated HRs, and estimated 95% confidence intervals were compared with those in the original publications. In case of apparent discrepancies, the extraction process was repeated.

### 2.4. Statistical Analysis, One- and Two-Stage Meta-Analysis with Individual Patient Data

Patient- and disease-specific characteristics of the included studies were recorded and compared using Pearson’s chi-squared test (two-sided). The IPD information of all time-to-rerupture data from all the included trials was pooled, and Kaplan–Meier curves of rerupture-free survival were constructed for the whole included patient cohort using the R package *Survminer* and *Survival* in R software version 4.3.1 (R Foundation for Statistical Computing, Vienna, Austria). The 1-, 6-, and 10-years rerupture-free survival probabilities were calculated. The hazard ratios (HRs) of each individual study as well as the pooled HR and corresponding 95% confidence intervals (CI) between clipped and coiled aSAH patients were estimated.

In the two-stage meta-analysis, we combined the estimated hazard ratios (HRs) and their corresponding 95% CI from individual studies using a random-effects model, employing the generic inverse variance method. The calculated HRs were logarithmically transformed (LN). For each study, the standard error (SE) was derived from the 95% CI using the formula: SE = (LN(upper CI limit) − LN(lower CI limit))/3.92, as outlined in the Cochrane Handbook for Systematic Reviews of Interventions, Version 6.4 [13]. Heterogeneity across the included studies was evaluated using I^2^ statistics, with a threshold of >50% displaying substantial heterogeneity [14]. The aggregated results were presented in forest plots utilizing Review Manager Web (RevMan Web Version 5.4.1 from The Cochrane Collaboration). Weight of the relative contribution of the included studies, based on the sample size, was considered in the estimation of the treatment effects. A significance level of *p* < 0.05 was considered statistically significant. To evaluate publication bias, we performed a two-step approach. Firstly, a visual assessment via funnel plots was performed. Funnel plots were created using the R package metafor. Secondly, Begg’s test was conducted to statistically evaluate the asymmetry of the data [15].

## 3. Results

### 3.1. Study Selection and Study Characteristics

After reviewing 240 Studies, two articles were included in the present meta-analysis (see Figure 1 displaying the PRISMA flow diagram).

The present meta-analysis includes 3153 patients. The Cerebral Aneurysm Rerupture After Treatment (CARAT) study is an ambidirectional cohort study combining both prospective and retrospective phases, whereas the International Subarachnoid Aneurysm Trial (ISAT) study constitutes a prospective randomized trial [16,17]. Both studies are multicenter trials. The CARAT study presents data from a single nation, specifically the USA, while the ISAT study is a multinational trial primarily conducted in the UK. In the CARAT study, 29.6% of patients underwent endovascular coiling, whereas in the ISAT trial, 50.1% of patients underwent endovascular coiling. Each of the included studies featured comprehensive follow-up schedules, with average durations ranging from 3.7 to 9 years. Both studies reported Kaplan–Meier charts and number at risk tables regarding the rerupture-free survival of endovascularly or neurosurgically treated ruptured aneurysms. The characteristics of the included studies are presented in Table 1.

### 3.2. Individual Patient Data Cohort Characteristics

Table 2 provides the distribution of patient- and disease-specific characteristics among the included studies [16,17]. Univariable analyses using Pearson’s chi-squared test (two-sided) were used to compare the proportions of the parameters sex, age, severity of aSAH, and aneurysm size. The *p*-values represent the results of the two-sided Pearson’s chi-squared test. Sixty-eight percent of the aSAH patients in the CARAT study were female, whereas in the ISAT study, 62.8% were female patients (*p* = 0.009). The mean and median ages of the aSAH patients were in the sixth decade of life in both studies. The CARAT study included a substantially higher proportion of poor-grade aSAH patients (World Federation of Neurosurgical Societies (WFNS) or Hunt and Hess grades IV–V) compared with the ISAT study (20.1% vs. 4.5%; *p* = 0.0001). Furthermore, 14.6% of the aSAH patients in the CARAT study had a ruptured aneurysm larger than 10 mm, and 7.2% had a ruptured aneurysm with a size >10 mm in the ISAT study (*p* = 0.0001).

The main focus of this meta-analysis is to compare the long-term outcomes of endovascular coiling versus neurosurgical clipping, specifically looking at the risk of “rerupture”. We pooled data from the included studies and categorized them based on whether patients were treated with endovascular coiling or neurosurgical clipping. We then compared various patient- and disease-specific characteristics between the two treatment modalities using the two-sided Pearson’s chi-squared test.

In the neurosurgical clipping group, 65.2% of the patients were female, compared to 64.1% in the endovascular coiling group (*p* = 0.57). Data on the clinical severity of aSAH were not available for 31 patients in the CARAT study. The distribution of poor-grade aSAH patients (WFNS or Hunt and Hess grades IV–V) was similar between the neurosurgical clipping and endovascular coiling groups (*p* = 0.27). Additionally, 6.2% of patients in the neurosurgical clipping group had a ruptured aneurysm larger than 10 mm, while 4.5% of patients in the endovascular coiling group had an aneurysm larger than 10 mm (*p* = 0.048) (Table 3).

### 3.3. Reconstructed Pooled Rerupture-Free Survival Curves and One-Stage Meta-Analysis of the Impact of Primary Aneurysm Treatment on Rerupture-Free Survival in aSAH

The reconstructed rerupture-free survival curves and side-by-side comparisons with the original curves were conducted. The estimated HRs and corresponding 95% CI of the included studies in the one-stage analysis are shown in Table 4.

All the reconstructed Kaplan–Meier charts and the published charts in the individual investigations were nearly identical, and the discrepancies in the patient number at risk tables were minor. The median (IQR) follow-up time of the reconstructed IPD was 6.1 years (0.5–11.7). The reconstructed rerupture-free survival curve for the pooled study cohort stratified using coiling and clipping is shown in Figure 2.

Eight reruptures (0.5%) were observed in the clipping group, whereas in the coiling group, 21 reruptures (1.5%) were found (Pearson’s chi-squared test (two-sided): *p* = 0.002). Seven reruptures of the clipping arm were observed in the first year, and only one case suffered from a rerupture in the seventh year after primary aSAH. In the coiling group, 10 reruptures occurred in the first year and further 11 reruptures were identified ranging from the second to the sixth year after primary aSAH. The one- and six-year rerupture-free survival rates for aSAH patients who underwent clipping were 99.5% and 99.4%, respectively. The group of aSAH patients who underwent coiling had one- and six-year rerupture-free survival rates of 99.2% and 98.1%. The log-rank test revealed a significant association between coiling and late rerupture compared with clipping (*p* = 0.01).

### 3.4. Two-Stage Meta-Analysis of Treatment Modalities Regarding Target Aneurysm Rerupture Based on Inidividual Patient Data

To assess the robustness of the results, a two-phase meta-analysis was carried out. Concerning rerupture-free survival, the pooled hazard ratio was 3.62 (95% CI: 1.21–10.86, *p* = 0.02) confirming the outcomes observed in the initial one-stage meta-analysis and affirming an association between coiling and risk of late rerupture of the initial target aneurysm after primary aSAH treatment. Conversely, clipping was associated with a reduced risk (HR (inverse hazard ratio and 95% CI): 0.28 (95% CI: 0.09–0.83, *p* = 0.02)) of rerupture of the initial target aneurysm causing recurrent aSAH. Figure 3 shows a forest plot displaying the results of the analysis. The investigation of rerupture-free survival demonstrated minimal heterogeneity across the studies (I^2^ = 37%, *p* = 0.21).

### 3.5. Publication Bias and Quality Evaluation

To ensure scientific accuracy, we performed the following three steps to investigate any potential publication bias: first, we performed an extensive literature search strategy; second, we used highly selective inclusion criteria; and finally, we investigated the publication bias of the included studies using a funnel plot and a statistical test. A funnel plot was created to visually examine the publication bias. Figure 4 displays a funnel plot of the two included studies. Begg’s test revealed no publication bias (*p* = 0.99).

The NIH-QAT tool was used to evaluate quality, resulting in favorable ratings for the included studies. The ratings for each of the 14 NIH-QAT domains can be found in Appendix A. Blinding of patients or physicians was not possible in the setting of interventions for treating ruptured aneurysms in both studies. Sample size justification was not intended in the case of the CARAT study, as it was constructed as an ambidirectional study.

## 4. Discussion

The present meta-analysis comparing clipping versus coiling for ruptured cerebral aneurysms analyzed two studies involving 3153 patients [16,17]. This study is the first to investigate longitudinal data using reconstructed individual patient data. The key findings are as follows: (1) The occurrence of rerupture in the initially ruptured target aneurysm is rare, with a rate of 0.92%; (2) Clipping is associated with a reduced risk of rerupture of the initial target aneurysm causing recurrent aSAH, with a hazard ratio of 0.28 (95% CI: 0.09–0.83). To our knowledge, this is an inaugural pooled meta-analysis elucidating the relationship between clipping and a reduced risk of rerupture in target ruptured aneurysms, utilizing reconstructed individual patient data to assess time-to-event data over an extended period.

Recurrent aSAH, while rare, poses a significant risk within the initial 10 years post-aSAH, approximately 22 times higher than that in similar populations [18]. While the occurrence of recurrent aSAH or rerupture of the target aneurysm is rare, the neurosurgical community is concerned about the potential loss of the initial clinical advantages of coiling over clipping in the long-term, particularly if there are high rates of rebleeding. In this meta-analysis, we discovered a rerupture rate of the initially ruptured target aneurysms after coil embolization of 1.5%, whereas after neurosurgical clipping, a rerupture rate of 0.5% was observed. The data presented in our meta-analysis provide reassurance. However, it is important to note that based on these data, the annual risk of rerupture of the initially treated target aneurysm is higher in patients who undergo endovascular coiling compared to those who undergo clipping. Nonetheless, the overall incidence of rerupture of target aneurysms still remains low. In the ISAT study, patients treated with endovascular coiling demonstrated a 7% absolute improvement in favorable clinical outcomes (mRS < 3) at the one-year mark compared to those treated with surgical clipping [19]. However, long-term endpoints, such as angiographic recurrence rate or rerupture causing recurrent aSAH, were found to be worse in patients who underwent coiling. Furthermore, it was shown that after excluding the data of patients who had poor outcomes due to death after rerupture before treatment in the clipping group in the ISAT study, the statistical significance for favorable outcome in the endovascular group was lost, even in the short-term follow-up [4]. In the published ten-year data of the BRAT study, retreatment rates were 20% for patients who underwent coiling vs. 0.8% in the clipping group [20]. Two deaths occurred due to subarachnoid hemorrhages during the 10-year follow-up of the BRAT study, both in patients who received endovascular coiling. One hemorrhage stemmed from an incidental coiled basilar artery aneurysm. No subarachnoid hemorrhages were observed during the 10-year follow-up in the clipping group of the BRAT study. Hence, there was only one rerupture from a target aneurysm in the BRAT study at the 10-year follow-up. However, the BRAT study could not be included in our analysis because no Kaplan–Meier curve analyses of target aneurysm rerupture have been published. Despite this higher retreatment rates, rerupture rates still remain significantly higher in the endovascular group compared to the clipping group. In an 18-year follow-up study of the UK cohort in the ISAT study, rerupture from the target aneurysm was observed in 1.2% of patients who underwent coiling, while the rate was 0.4% in those who underwent clipping [21]. However, ongoing criticism has been directed towards the findings of the ISAT study, mainly attributed to concerns about selection bias. Within the study cohort, 88% of patients had a favorable functional grade (classified as per the WNFS classification I and II) upon enrollment, and 95% of the aneurysms were situated in the anterior cerebral circulation, with 90% being smaller than 10 mm in size [21]. The CARAT study reported similar results regarding the rerupture of the target aneurysm, with more frequent rates in those treated via endovascular embolization [16]. The CARAT study was conducted with an observational design, including all aneurysms treated during the specified time period. Therefore, this study design is more susceptible to introducing significant differences in baseline demographics between the aneurysm occlusion therapies (clipping and coiling). However, after pooling the data from both studies and dichotomizing the present study cohort using clipping and coiling, no significant differences were observed, except for a higher frequency of aneurysms ≥10 mm in the clipping arm. The findings of our meta-analysis were also supported by another conventional meta-analysis of dichotomous data by Li et al. [22], which found that coiling is associated with an increased risk of rebleeding of the treated aneurysm (OR:2.43, 95% CI: 1.88–3.13). However, the present investigation represents a longitudinal time-to-event data meta-analysis.

As far as the long-term outcome is concerned, the risk of rerupture might be considered in the preoperative decision-making process despite the known better outcome after coiling at one year after aSAH [22]. This finding might be of importance because at 10 years after aSAH, the proportions of a good mRS score (0–2) did not differ among those who underwent clipping or coiling in the UK cohort of the ISAT trial [21]. Ruptured aneurysms being coiled are associated with a higher long-term risk of rerupture. The ARETA trial revealed that around 20% of patients experience either the reopening of the aneurysms or its neck after endovascular therapy, leading to the need for retreatment in approximately half of these cases [23]. Fortunately, aneurysm rerupture of the target aneurysm remains a rare event. However, this finding further highlights the need for a consensus or established guideline regarding retreatment after coiling for ruptured aneurysms. The risk of early rebleeding after therapy is known to be strongly associated with the extent of aneurysm occlusion according to the CARAT study [24]. Zhu et al. [25] revealed that clipping increases the incidence of complete aneurysmal occlusion by 33% compared to coiling. A literature review of 26 reports showed that delayed aneurysm rupture after coiling of unruptured aneurysms occurs in most cases when there is a remaining portion of the aneurysm, although it is also possible in cases where there is a remaining portion of the neck [26]. Hence, the present meta-analysis of time-to-event data supports the strategy that growing amenable remnants or persistent amenable remnants of coiled aneurysms should be subjected to retreatment in case of patient’s good physical status and long-life expectancy in order to prevent the potential risk of experiencing aneurysm rerupture. This “aggressive” strategy is also supported by the retrospective study of Mendenhall et al. [27], who analyzed 214 coil-embolized previously ruptured cerebral aneurysm, with retreatment (indicated for incomplete initial coiling, coil compaction, and aneurysmal dilatation) performed in 46 cases. The rate of rerupture was 0.9% in the study by Mendenhall et al. [27], which is lower than the reported rerupture rates after coiling compared with our meta-analysis and literature reporting rates between 2.3 and 3.0% [22]. However, it should be noted that the present study does not show a direct statistical correlation between remnants and late rerupture. The present meta-analysis is based on endovascular treatment results without the use of new additions, such as stents, Woven EndoBridge (WEB) device, or flow diverters, which enable an expansion of the indications of endovascular therapy to include wide-necked or giant aneurysms [28,29,30]. To date, it is unclear whether this wider spectrum of patients and aneurysms, now being considered for endovascular therapy, also benefit from the positive results regarding the one-year clinical outcomes such as survival without dependency, as described in the ISAT study [31]. The ongoing ISAT II study (Trial Number: NCT01668563) is estimated to be completed in June 2024 and might allow a further stratification of the endpoint late rerupture of target aneurysms using various treatment modalities (clipping, coiling, coiling with stent, WEB device, and flow diverter).

### Strengths and Limitations

This meta-analysis possesses several strengths. One of the notable strengths is the utilization of the most appropriate methodology to analyze the data, which involves individual patient data. We made sure to include publications of studies reporting the rerupture rates of the initially treated target aneurysms. By examining the Kaplan–Meier curves and number at risk tables within these publications, we were able to extract the rerupture-free survival data of each individual participant for both trials. Performing a meta-analysis of individual patient time-to-event data allowed us to produce more reliable and robust results compared to traditional aggregate data meta-analyses. Moreover, the validity of our findings was enhanced through two-stage meta-analyses. Additionally, it is worth mentioning that there was no substantial heterogeneity for rerupture-free survival among the patient characteristics. Furthermore, our pooled coiling and clipping IPD cohorts did not differ regarding the clinical severity of aSAH patients, which impedes a potential significant amount of shorter follow-up in one treatment arm because of poor-grade aSAH.

This meta-analysis is not without its limitations. One notable limitation is that the individual patient data obtained from Kaplan–Meier curves only provided rerupture-free survival data at the patient level, without including other important variables such as degree of occlusion, hypertension, localization and treatment modality of the ruptured aneurysm (e.g., subgroup analysis of ruptured anterior communicating artery aneurysms: clipping vs. coiling regarding long-term rerupture risk), and smoking status. However, we could compare the cohorts of coiling and clipping regarding the proportions of aneurysm size, sex, and baseline WFNS grades. Unfortunately, the currently available individual patient data did not enable us to perform subgroup analyses with longitudinal data based on factors such as the degree of occlusion. This limitation is further compounded by the lack of sufficient study-level data to perform such analyses. Therefore, it is necessary to conduct a meta-analysis of individual patient data that includes baseline characteristics obtained directly from the authors of each study in order to address these issues.

## 5. Conclusions

The present meta-analysis of large-scale studies shows a generally low overall rerupture rate, but rerupture occurs significantly more often in aSAH patients who underwent endovascular coiling compared with clipping of the initially ruptured target aneurysm. Despite the known advantages of coiling regarding the one-year clinical outcome, late rerupture of the target aneurysm might also be considered in the pretreatment decision-making process of aSAH patients.

## Figures and Tables

**Figure 1 jcm-13-01778-f001:**
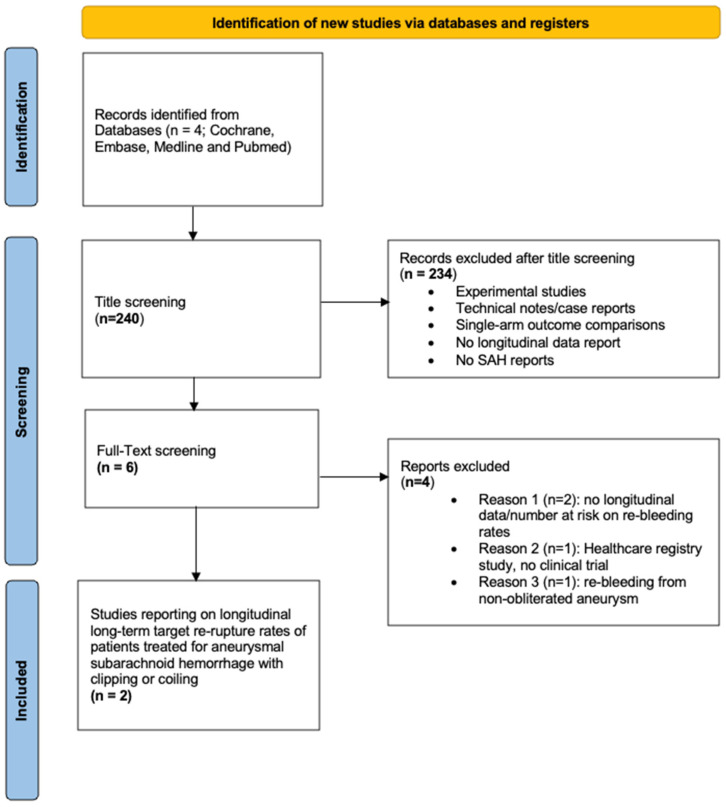
PRISMA flow diagram illustrating the search strategy.

**Figure 2 jcm-13-01778-f002:**
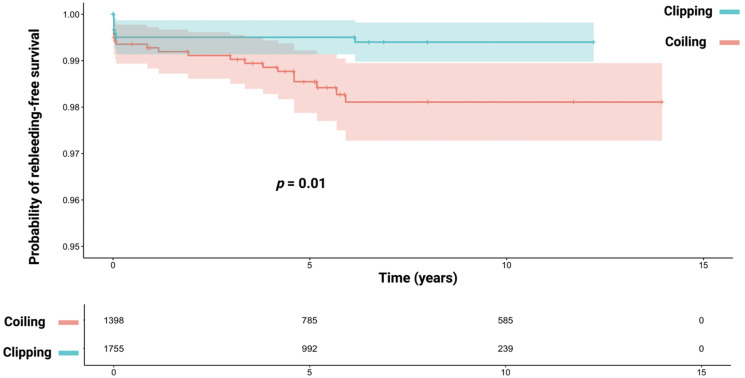
Kaplan–Meier chart displaying probabilities of rerupture-free survival stratified using clipping (turquoise) and coiling (red). The shadowed areas surrounding the curves display the confidence intervals. The log-rank test (*p* = 0.01) shows a significant association between coiling and late rerupture. The patient number at risk table is given below the Kaplan–Meier chart.

**Figure 3 jcm-13-01778-f003:**
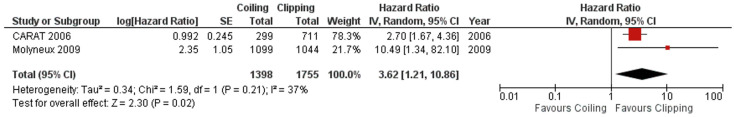
Forest plot displaying log(hazard ratio(HR)), standard error, HR estimates, and 95% CI estimates for rerupture-free survival in the included studies [16,17] evaluating coiling compared to clipping in aSAH patients. X-axis locations of squares display the natural logarithm of the hazard ratio; the bigger the square, the greater the weight of the study. The diamond corresponds to the logHR of the pooled data.

**Figure 4 jcm-13-01778-f004:**
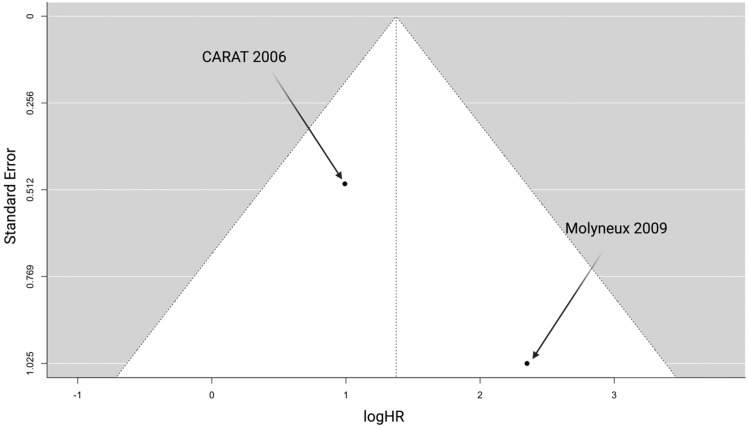
Funnel plot for the endpoint rerupture-free survival [16,17].

**Table 1 jcm-13-01778-t001:** Characteristics of the included studies.

Source	Country	Study Interval	Study Design	Total No.	Endovascular Coiling, No. (%)	Neurosurgical Clipping, No. (%)	Follow-Up
CARAT Investigators et al., 2006 [16]	United States of America	1 January 1996–31 December 1998	Ambidirectional multicentric Cohort study	1010	299 (29.6%)	711 (70.4%)	Clipping: 4.4 years (mean) Coiling: 3.7 years (mean)
Molyneux et al., 2009 [17]	United Kingdom, several European countries, Australia, Canada, and United States of America	1997–May 2002	Prospective randomized multicentric study	2143	1073 (50.1%)	1070 (49.9%)	9 years (mean)

**Table 2 jcm-13-01778-t002:** Distribution of patient- and disease-specific characteristics of aSAH patients in the included studies.

Parameters	CARAT Investigators et al. [16]	Molyneux et al. [17]	*p* *
Sex Female Male	696/1010 (68.9%) 314/1010 (31.1%)	1345/2143 (62.8%) 798/2143 (37.2%)	0.009
Age (in years)	Coiling: 58.5 (mean) (SD:15.1) Clipping: 53.5 (mean) (SD:13.8)	Coiling: 52.0 (median) (IQR:44.0–60.0) Clipping: 52.0 (median) (IQR: 43–60)	NA
Poor-grade aSAH (WFNS or Hunt and Hess grades IV–V)	203/1010 (20.1%)	94/2112 (4.5%) (unavailable WFNS data in 31 patients)	0.0001
Aneurysm size >10 mm	147/1010 (14.6%)	155/2143 (7.2%)	0.0001

Abbreviations: NA = Not applicable, IQR = Interquartile range, SD = Standard deviation, WFNS = World Federation of Neurosurgical Societies. *p* * value derived from two-sided Pearson’s chi-squared test.

**Table 3 jcm-13-01778-t003:** Distribution of baseline patient- and disease-specific characteristics of aSAH patients in the included studies for the pooled clipping and coiling groups (lost to follow-up in both arms because of missing rebleeding data in CARAT et al. [16] and in Molyneux et al. [17]).

Parameters	Clipping	Coiling	*p* *
Sex Female Male	1161/1781 (65.2%) 620/1781 (34.8%)	880/1372 (64.1%) 492/1372(35.9%)	0.57
Poor-grade aSAH (WFNS or Hunt and Hess grades IV–V)	177/1765 (10.0%) (Not available in 1 patient in CARAT Investigators et al. [16] and in 16 patients of Molyneux et al. [17])	120/1357 (8.8%) (Not available in 5 patients in CARAT Investigators et al. [16] and in 15 patients of Molyneux et al. [17])	0.27
Aneurysm size >10 mm	110/1781 (6.2%)	62/1372 (4.5%)	0.048

Abbreviation: WFNS = World Federation of Neurosurgical Societies; *p* * value derived from two-sided Pearson’s chi-squared test.

**Table 4 jcm-13-01778-t004:** Estimated hazard ratios (HR) and corresponding 95% confidence intervals regarding rerupture-free survival in the one-stage meta-analysis.

Endpoint	Study, Year	Estimated HR (95% CI)
Rerupture-free survival	CARAT Investigators et al., 2006 [16]	2.70 (1.03–7.05)
	Molyneux et al., 2009 [17]	10.52 (1.34–82.50)

Abbreviations: CI = Confidence Interval; HR = Hazard Ratio.

## Data Availability

The raw data supporting the conclusions of this article will be made available by the authors on request.

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
