# Peer review of "A Long-Term Comparative Analysis of Endovascular Coiling and Clipping for Ruptured Cerebral Aneurysms: An Individual Patient-Level Meta-Analysis Assessing Rerupture Rates"

_jcm, 2024, doi:10.3390/jcm13061778_

Round 1
Reviewer 1 Report
Comments and Suggestions for Authors
The review entitled “A Long-Term Comparative Analysis of Endovascular Coiling and Clipping for Ruptured Cerebral Aneurysms: An Individual Patient-Level Meta-Analysis Assessing Re-Rupture Rates” examines long-term re-rupture rates in aSAH patients treated with coiling versus clipping. The study found that the re-rupture rate was 0.5% for clipping and 1.5% for coiling and recommends that this re-rupture plausibility should be considered before finalizing the treatment modality for aSAH.
Suggestions/Limitations
1. Include more recent studies (line 31)
2. The review results are based on only two selected studies conducted previously
3. What additional information is delivered to the scientific community from this study compared to the study by Li et al (ref 21) and other similar studies mentioned below
Xia ZW, Liu XM, Wang JY, Cao H, Chen FH, Huang J, Li QZ, Fan SS, Jiang B, Chen ZG, Cheng Q. Coiling Is Not Superior to Clipping in Patients with High-Grade Aneurysmal Subarachnoid Hemorrhage: Systematic Review and Meta-Analysis. World Neurosurg. 2017 Feb;98:411-420. doi: 10.1016/j.wneu.2016.11.032. Epub 2016 Nov 17. PMID: 27867126.
Shao B, Wang J, Chen Y, He X, Chen H, Peng Y, Yang P, Duan H, Yang F, Teng L. Clipping versus Coiling for Ruptured Intracranial Aneurysms: A Meta-Analysis of Randomized Controlled Trials. World Neurosurg. 2019 Jul;127:e353-e365. doi: 10.1016/j.wneu.2019.03.123. Epub 2019 Mar 27. PMID: 30928577.
Lindgren A, Vergouwen MD, van der Schaaf I, Algra A, Wermer M, Clarke MJ, Rinkel GJ. Endovascular coiling versus neurosurgical clipping for people with aneurysmal subarachnoid haemorrhage. Cochrane Database Syst Rev. 2018 Aug 15;8(8):CD003085. doi: 10.1002/14651858.CD003085.pub3. PMID: 30110521; PMCID: PMC6513627.
Falk Delgado A, Andersson T, Falk Delgado A. Clinical outcome after surgical clipping or endovascular coiling for cerebral aneurysms: a pragmatic meta-analysis of randomized and non-randomized trials with short- and long-term follow-up. J Neurointerv Surg. 2017 Mar;9(3):264-277. doi: 10.1136/neurintsurg-2016-012292. Epub 2016 Apr 6. PMID: 27053705.
4. Discussion should include more related studies
5. Hence, the present 308 study supports the strategy that evolving remnants and remnants of aneurysms should 309 be subjected to retreatment in case of patient´s good functionality and life expectancy due 310 to the potential risk of experiencing re-rupture (Line 308-310). Explain clearly.
Author Response
Dear Reviewer,
Thank you for reading our manuscript and critically reviewing it, which will help us improve it to a better scientific level and make it more understandable to the readership.
Consequently, we have submitted a revised version of the manuscript containing all the changes to be visible (highlighted in red).
At the following, the remarks mentioned by the reviewers will be discussed:
“Include more recent studies”
We absolutely agree that the generic beginning of an introduction describing the prognosis (e.g., mortality) of a disease should cite up-to-date research. Hence, we revised this sentence and cited the recent study by Scibilia et al. [1] which analyzed 149 patients with poor-grade aneurysmal subarachnoid hemorrhage (WFNS grades IV & V) and found that 1-month mortality rate was 21%.
References
- Scibilia, A.; Rustici, A.; Linari, M.; Zenesini, C.; Belott,i L.M.B.; Dall'Olio, M.; Princiotta, C.; Cuoci, A.; Aspide, R.; Migliorino, E.; Moneti, M.; Sturiale, C.; Castioni, C.A.; Conti, A.; Bortolotti, C.; Cirillo, L. Factors affecting 30-day mortality in poor-grade aneurysmal subarachnoid hemorrhage: a 10-year single-center experience. Front Neurol 2024, 15, 1286862. doi: 10.3389/fneur.2024.1286862.
“The review results are based on only two selected studies conducted previously. What additional information is delivered to the scientific community from this study compared to the study by Li et al (ref 21) and other similar studies mentioned below.”
The reviewer is absolutely right that the results are only based on two studies. However, only these two studies in the literature provide longitudinal time-to event data of target aneurysm rerupture and we think the pooling of longitudinal follow-up data is needed to understand how long follow-up imaging is needed and when retreatment should be initiated. A conventional meta-analysis comparing the efficacy of clipping versus coiling by pooling only the dichotomized data (target aneurysm rerupture: yes or no) is inappropriate to address this issue. Several meta-analyses [1-4] have endeavored to compare the efficacy of clipping versus coiling in the treatment of aneurysms by pooling dichotomous data. However, all of these conventional meta-analyses combined dichotomous data, a practice discouraged by the Cochrane handbook [5], potentially yielding not entirely appropriate conclusions. Conversely, time-to-event data have been identified as the most suitable for time-to-event data meta-analyses, with individual participant data (IPD) offering significant advantages for such analyses [6]. However, these mentioned meta-analyses were published between 2017-2019 and this novel method by Liu et al. [7] to reconstruct individual patient data was not possible for the previous meta-analyses. Consequently, we conducted this IPD meta-analysis utilizing time-to-event data (target aneurysm rerupture stratified by aneurysm occlusion technique) to examine the long-term target aneurysm rerupture outcomes following clipping or coiling in adult patients after aneurysmal subarachnoid hemorrhage.
References
- Xia ZW, Liu XM, Wang JY, Cao H, Chen FH, Huang J, Li QZ, Fan SS, Jiang B, Chen ZG, Cheng Q. Coiling Is Not Superior to Clipping in Patients with High-Grade Aneurysmal Subarachnoid Hemorrhage: Systematic Review and Meta-Analysis. World Neurosurg. 2017 Feb;98:411-420. doi: 10.1016/j.wneu.2016.11.032. Epub 2016 Nov 17. PMID: 27867126.
- Shao B, Wang J, Chen Y, He X, Chen H, Peng Y, Yang P, Duan H, Yang F, Teng L. Clipping versus Coiling for Ruptured Intracranial Aneurysms: A Meta-Analysis of Randomized Controlled Trials. World Neurosurg. 2019 Jul;127:e353-e365. doi: 10.1016/j.wneu.2019.03.123. Epub 2019 Mar 27. PMID: 30928577.
- Lindgren A, Vergouwen MD, van der Schaaf I, Algra A, Wermer M, Clarke MJ, Rinkel GJ. Endovascular coiling versus neurosurgical clipping for people with aneurysmal subarachnoid haemorrhage. Cochrane Database Syst Rev. 2018 Aug 15;8(8):CD003085. doi: 10.1002/14651858.CD003085.pub3. PMID: 30110521; PMCID: PMC6513627.
- Falk Delgado A, Andersson T, Falk Delgado A. Clinical outcome after surgical clipping or endovascular coiling for cerebral aneurysms: a pragmatic meta-analysis of randomized and non-randomized trials with short- and long-term follow-up. J Neurointerv Surg. 2017 Mar;9(3):264-277. doi: 10.1136/neurintsurg-2016-012292. Epub 2016 Apr 6. PMID: 27053705.
- Chapter 6: Choosing effect measures and computing estimates of effect. In: Higgins JPT, Thomas J, Chandler J, et al. eds. Cochrane Handbook for Systematic Reviews of Interventions version 6.4. Accessed March 7th. https://training.cochrane.org/handbook/current/chapter-06
- Riley RD, Lambert PC, Abo-Zaid G. Meta-analysis of individual participant data: rationale, conduct, and reporting. BMJ. 2010 Feb 5;340:c221. doi: 10.1136/bmj.c221.
- Liu, N.; Zhou, Y.; Lee, J.J. IPDfromKM: reconstruct individual patient data from published Kaplan-Meier survival curves. BMC Med Res Methodol 2021, 21(1), 111. doi: 10.1186/s12874-021-01308-8.
“Discussion should include more related studies”
The reviewer is absolutely right that the discussion benefits from debating more related studies. Fortunately, the occurrence of aneurysm rerupture in the target aneurysm remains an uncommon event. However, the present observations underscores the urgency for consensus or established guidelines concerning retreatment following coil-embolized ruptured aneurysms with remnants. As per the CARAT study [1], the risk of early re-bleeding post-therapy is closely linked to the degree of aneurysm occlusion. Zhu et al. [2] demonstrated a 33% increase in complete aneurysmal occlusion incidence with clipping compared to coiling. A comprehensive review of 26 reports revealed that delayed aneurysm rupture after coiling of unruptured aneurysms primarily occurs when a portion of the aneurysm remains, though it may also transpire when a segment of the neck remains [3]. Consequently, the present meta-analysis of time-to-event data advocates for a proactive approach wherein growing or persistent amenable remnants of coiled aneurysms should be retreated in cases of patients with good physical status and long-life expectancy to mitigate the risk of aneurysm re-rupture. This approach is further supported by Mendenhall et al.'s retrospective study [4], which examined 214 previously ruptured cerebral aneurysms treated with coil embolization; retreatment (indicated for incomplete initial coiling, coil compaction, aneurysmal dilatation) was performed in 46 cases. The rerupture rate in Mendenhall et al.'s study [4] was 0.9%, which is lower than the rates reported in our meta-analysis and the literature, ranging between 2.3-3.0% [5]. However, it's important to note that the present study doesn't establish a direct statistical correlation between remnants and late rerupture. Moreover, this meta-analysis is based on endovascular treatment outcomes without the incorporation of newer technologies such as stents, Woven EndoBridge (WEB) Device, or flow-diverters, which broaden the scope of endovascular therapy to encompass wide-necked or giant aneurysms [6-8].
References
- Johnston, S.C.; Dowd, C.F.; Higashida, R.T.; Lawton, M.T.; Duckwiler, G.R.; Gress, D.R.; CARAT Investigators. Predictors of rehemorrhage after treatment of ruptured intracranial aneurysms: the Cerebral Aneurysm Rerupture After Treatment (CARAT) study. Stroke 2008, 39(1), 120-5. doi: 10.1161/STROKEAHA.107.495747.
- Zhu, W.; Ling, X.; Petersen, J.D.; Liu, J.; Xiao, A.; Huang, J. Clipping versus coiling for aneurysmal subarachnoid hemorrhage: a systematic review and meta-analysis of prospective studies. Neurosurg Rev 2022, 45(2), 1291-1302. doi: 10.1007/s10143-021-01704-0.
- Tsurumi, A.; Tsurumi, Y.; Negoro, M.; Tsugane, S.; Ryuge, M.; Susaki, N.; Fukuoka, T.; Miyachi, S. Delayed rupture of a basilar artery aneurysm treated with coils: case report and review of the literature. J Neuroradiol 2013, 40(1), 54-61. doi: 10.1016/j.neurad.2012.08.005.
- Mendenhall, S.K.; Shapiro, S.A.; Cohen-Gadol, A.A.; Sahlein, D.H. Endovascular Retreatment of Previously Ruptured Coiled Cerebral Aneurysm Remnants Significantly Reduces Rebleed Rate. World Neurosurg 2021, 147, e382-e387. doi: 10.1016/j.wneu.2020.12.063.
- Li, H.; Pan, R.; Wang, H.; Rong, X.; Yin, Z.; Milgrom, D.P.; Shi, X.; Tang, Y.; Peng, Y. Clipping versus coiling for ruptured intracranial aneurysms: a systematic review and meta-analysis. Stroke 2013, 44(1), 29-37. doi: 10.1161/STROKEAHA.112.663559.
- Bae, H.J.; Park, Y.K.; Cho, D.Y.; Choi, J.H.; Kim, B.S.; Shin, Y.S. Predictors of the Effects of Flow Diversion in Very Large and Giant Aneurysms. AJNR Am J Neuroradiol 2021, 42(6), 1099-1103. doi: 10.3174/ajnr.A7085.
- Cagnazzo, F.; Ahmed, R.; Dargazanli, C.; Lefevre, P.H.; Gascou, G.; Derraz, I.; Kalmanovich, S.A.; Riquelme, C.; Bonafe, A.; Costalat, V. Treatment of Wide-Neck Intracranial Aneurysms with the Woven EndoBridge Device Associated with Stenting: A Single-Center Experience. AJNR Am J Neuroradiol 2019, 40(5), 820-826. doi: 10.3174/ajnr.A6032.
- Park, K.Y.; Jang, C.K.; Lee, J.W.; Kim, D.J.; Kim, B.M.; Chung, J. Preliminary experience of stent-assisted coiling of wide-necked intracranial aneurysms with a single microcatheter. BMC Neurol 2019, 19(1), 245. doi: 10.1186/s12883-019-1470-8.
“Hence, the present 308 study supports the strategy that evolving remnants and remnants of aneurysms should 309 be subjected to retreatment in case of patient´s good functionality and life expectancy due 310 to the potential risk of experiencing re-rupture (Line 308-310). Explain clearly.”
We agree with the reviewer that this sentence may benefit from a revision to enhance the understanding. Therefore, we have revised the lines 309-313 in the section discussion: “Hence, the present meta-analysis of time-to-event data regarding target aneurysm rerupture stratified by treatment modality (coiling or clipping) supports the strategy that progressive amenable remnants or stable amenable remnants of aneurysms should be subjected to retreatment in case of patient´s good physical status and long life expectancy in order to prevent the potential risk of experiencing aneurysm re-rupture.”
Reviewer 2 Report
Comments and Suggestions for Authors
1. Minor typo in the abstract 'numer'
2. In the flow chart (Fig. 1) it is not clear to me why 234 records were excluded. Reasons must be provided.
3. In the flow chart (Fig. 1) numbers are missing regarding the four excluded reports that were assessed for eligibility
4. Two studies that were included differed significantly in the most important aspects of the subject and aneurysm characteristics (Hunt Hess, aneurysm size, sex), which raises a question whether meta-analysis of such studies is appropriate.
5. Although a meta-analysis requires at least 2 studies, in order to reach a statistical power higher than the power of the individual studies included in a meta-analysis, two of the included studies should be of low heterogeneity. It is not the case here.
6. Because of the abovementioned, the generalizability of the findings is low and should be clearly stated in the limitations.
Comments on the Quality of English LanguageMinor typos.
Author Response
Dear Reviewer,
Thank you for reading our manuscript and critically reviewing it, which will help us improve it to a better scientific level and make it more understandable to the readership.
Consequently, we have submitted a revised version of the manuscript containing all the changes to be visible (highlighted in red).
At the following, the remarks mentioned by the reviewers will be discussed:
- Minor typo in the abstract 'numer'
Thank you for the recommendation. The manuscript underwent a professional language editing by a native speaker.
- In the flow chart (Fig. 1) it is not clear to me why 234 records were excluded. Reasons must be provided.
- In the flow chart (Fig. 1) numbers are missing regarding the four excluded reports that were assessed for eligibility
We absolutely agree with the reviewer regarding the remark concerning the PRISMA-flow-chart. Therefore, we added the reasons with the corresponding numbers of excluded manuscripts to the PRISMA-Flowchart diagram. Hence, we have inserted the revised Figure 1 to the manuscript in the section „3.1 Study Selection and Study Characteristics. The four excluded studies during the full-text screening are the following:
- Zhang J, Lo YL, Li MC, Yu YH, Wu SY. Risk of Re-Rupture, Vasospasm, or Re-Stroke after Clipping or Coiling of Ruptured Intracranial Aneurysms: Long-Term Follow-Up with a Propensity Score-Matched, Population-Based Cohort Study. J Pers Med. 2021 Nov 16;11(11):1209. doi: 10.3390/jpm11111209.
- Spetzler RF, McDougall CG, Zabramski JM, Albuquerque FC, Hills NK, Nakaji P, Karis JP, Wallace RC. Ten-year analysis of saccular aneurysms in the Barrow Ruptured Aneurysm Trial. J Neurosurg. 2019 Mar 8;132(3):771-776. doi: 10.3171/2018.8.JNS181846.
- Koivisto T, Vanninen R, Hurskainen H, Saari T, Hernesniemi J, Vapalahti M. Outcomes of early endovascular versus surgical treatment of ruptured cerebral aneurysms. A prospective randomized study. Stroke. 2000 Oct;31(10):2369-77. doi: 10.1161/01.str.31.10.2369.
- Brilstra EH, Rinkel GJ, Algra A, van Gijn J. Rebleeding, secondary ischemia, and timing of operation in patients with subarachnoid hemorrhage. Neurology. 2000 Dec 12;55(11):1656-60. doi: 10.1212/wnl.55.11.1656.
- Two studies that were included differed significantly in the most important aspects of the subject and aneurysm characteristics (Hunt Hess, aneurysm size, sex), which raises a question whether meta-analysis of such studies is appropriate.
Thank you for pointing out the study-discrepancies. The review is absolutely right that this could be a problem in a conventional meta-analysis investigating only dichotomous data of the individual included studies. Hence, we performed an indivdual patient-data meta-analysis to address this issue of conventional meta-analysis. We have pooled the individual patient characteristics of both studies stratified by coiling and clipping. The table 2 shows that the patient characteristics regarding sex and Hunt/Hess/WFNS grade do not differ.
However, there is only an increased proportion of patients with large aneurysms (>10 mm) among the clipping cohort. The detailed information on this is provided in Table 3.
Furthermore, we have revised the paragraph describing the results in order to asses the importance of this comparison and facilitate the understanding of the paragraph.
References
- Xia ZW, Liu XM, Wang JY, Cao H, Chen FH, Huang J, Li QZ, Fan SS, Jiang B, Chen ZG, Cheng Q. Coiling Is Not Superior to Clipping in Patients with High-Grade Aneurysmal Subarachnoid Hemorrhage: Systematic Review and Meta-Analysis. World Neurosurg. 2017 Feb;98:411-420. doi: 10.1016/j.wneu.2016.11.032. Epub 2016 Nov 17. PMID: 27867126.
- Shao B, Wang J, Chen Y, He X, Chen H, Peng Y, Yang P, Duan H, Yang F, Teng L. Clipping versus Coiling for Ruptured Intracranial Aneurysms: A Meta-Analysis of Randomized Controlled Trials. World Neurosurg. 2019 Jul;127:e353-e365. doi: 10.1016/j.wneu.2019.03.123. Epub 2019 Mar 27. PMID: 30928577.
- Lindgren A, Vergouwen MD, van der Schaaf I, Algra A, Wermer M, Clarke MJ, Rinkel GJ. Endovascular coiling versus neurosurgical clipping for people with aneurysmal subarachnoid haemorrhage. Cochrane Database Syst Rev. 2018 Aug 15;8(8):CD003085. doi: 10.1002/14651858.CD003085.pub3. PMID: 30110521; PMCID: PMC6513627.
- Falk Delgado A, Andersson T, Falk Delgado A. Clinical outcome after surgical clipping or endovascular coiling for cerebral aneurysms: a pragmatic meta-analysis of randomized and non-randomized trials with short- and long-term follow-up. J Neurointerv Surg. 2017 Mar;9(3):264-277. doi: 10.1136/neurintsurg-2016-012292. Epub 2016 Apr 6. PMID: 27053705.
- Chapter 6: Choosing effect measures and computing estimates of effect. In: Higgins JPT, Thomas J, Chandler J, et al. eds. Cochrane Handbook for Systematic Reviews of Interventions version 6.4. Accessed March 7th. https://training.cochrane.org/handbook/current/chapter-06
- Riley RD, Lambert PC, Abo-Zaid G. Meta-analysis of individual participant data: rationale, conduct, and reporting. BMJ. 2010 Feb 5;340:c221. doi: 10.1136/bmj.c221.
- Liu, N.; Zhou, Y.; Lee, J.J. IPDfromKM: reconstruct individual patient data from published Kaplan-Meier survival curves. BMC Med Res Methodol 2021, 21(1), 111. doi: 10.1186/s12874-021-01308-8.
- Although a meta-analysis requires at least 2 studies, in order to reach a statistical power higher than the power of the individual studies included in a meta-analysis, two of the included studies should be of low heterogeneity. It is not the case here.
The reviewer is absolutely right that the results are only based on two studies. However, only these two studies in the literature provide longitudinal time-to event data of target aneurysm rerupture and we think the pooling of longitudinal follow-up data is needed to understand how long follow-up imaging is needed and when retreatment should be initiated. A conventional meta-analysis comparing the efficacy of clipping versus coiling by pooling only the dichotomized data (target aneurysm rerupture: yes or no) is inappropriate to address this issue. Several meta-analyses [1-4] have endeavored to compare the efficacy of clipping versus coiling in the treatment of aneurysms by pooling dichotomous data. However, all of these conventional meta-analyses combined dichotomous data, a practice discouraged by the Cochrane handbook [5], potentially yielding not entirely appropriate conclusions. Conversely, time-to-event data have been identified as the most suitable for time-to-event data meta-analyses, with individual participant data (IPD) offering significant advantages for such analyses [6]. However, these mentioned meta-analyses were published between 2017-2019 and this novel method by Liu et al. [7] to reconstruct individual patient data was not possible for the previous meta-analyses. Consequently, we conducted this IPD meta-analysis utilizing time-to-event data (target aneurysm rerupture stratified by aneurysm occlusion technique) to examine the long-term target aneurysm rerupture outcomes following clipping or coiling in adult patients after aneurysmal subarachnoid hemorrhage.
References
- Xia ZW, Liu XM, Wang JY, Cao H, Chen FH, Huang J, Li QZ, Fan SS, Jiang B, Chen ZG, Cheng Q. Coiling Is Not Superior to Clipping in Patients with High-Grade Aneurysmal Subarachnoid Hemorrhage: Systematic Review and Meta-Analysis. World Neurosurg. 2017 Feb;98:411-420. doi: 10.1016/j.wneu.2016.11.032. Epub 2016 Nov 17. PMID: 27867126.
- Shao B, Wang J, Chen Y, He X, Chen H, Peng Y, Yang P, Duan H, Yang F, Teng L. Clipping versus Coiling for Ruptured Intracranial Aneurysms: A Meta-Analysis of Randomized Controlled Trials. World Neurosurg. 2019 Jul;127:e353-e365. doi: 10.1016/j.wneu.2019.03.123. Epub 2019 Mar 27. PMID: 30928577.
- Lindgren A, Vergouwen MD, van der Schaaf I, Algra A, Wermer M, Clarke MJ, Rinkel GJ. Endovascular coiling versus neurosurgical clipping for people with aneurysmal subarachnoid haemorrhage. Cochrane Database Syst Rev. 2018 Aug 15;8(8):CD003085. doi: 10.1002/14651858.CD003085.pub3. PMID: 30110521; PMCID: PMC6513627.
- Falk Delgado A, Andersson T, Falk Delgado A. Clinical outcome after surgical clipping or endovascular coiling for cerebral aneurysms: a pragmatic meta-analysis of randomized and non-randomized trials with short- and long-term follow-up. J Neurointerv Surg. 2017 Mar;9(3):264-277. doi: 10.1136/neurintsurg-2016-012292. Epub 2016 Apr 6. PMID: 27053705.
- Chapter 6: Choosing effect measures and computing estimates of effect. In: Higgins JPT, Thomas J, Chandler J, et al. eds. Cochrane Handbook for Systematic Reviews of Interventions version 6.4. Accessed March 7th. https://training.cochrane.org/handbook/current/chapter-06
- Riley RD, Lambert PC, Abo-Zaid G. Meta-analysis of individual participant data: rationale, conduct, and reporting. BMJ. 2010 Feb 5;340:c221. doi: 10.1136/bmj.c221.
- Liu, N.; Zhou, Y.; Lee, J.J. IPDfromKM: reconstruct individual patient data from published Kaplan-Meier survival curves. BMC Med Res Methodol 2021, 21(1), 111. doi: 10.1186/s12874-021-01308-8.
Round 2
Reviewer 2 Report
Comments and Suggestions for Authors
The authors have appropriately addressed my comments. I endorse acceptance.
Comments on the Quality of English LanguageAbbreviation IPD is explained twice in abstract.